# The integration of technology into a home-based visuo-cognitive training intervention for people with Parkinson's: Is the future digital?

**Julia Das**[1,2], **Gill Barry**[1], **Richard Walker**[2], **Rodrigo Vitorio**[1], **Rosie Morris**[1,2], **Samuel Stuart**[1,2]*

1 Department of Sport, Exercise and Rehabilitation, Northumbria University, Newcastle upon Tyne, United Kingdom, 2 Northumbria Healthcare NHS Foundation Trust, North Tyneside General Hospital, Newcastle upon Tyne, United Kingdom

* samuel2.stuart@northumbria.ac.uk

**Data Availability Statement:** All relevant data are within the paper and its supporting information files. Any further excerpts could be made available upon request but only those that have been

## Abstract

### Background

Mobile applications and technology (e.g., stroboscopic glasses) are increasingly being used to deliver combined visual and cognitive (termed visuo-cognitive) training that replaces standard pen and paper-based interventions. These 'technological visuo-cognitive training' (TVT) interventions could help address the complex problems associated with visuo-cognitive dysfunction in people with long term neurological conditions such as Parkinson's disease. As data emerges to support the effectiveness of these technologies, patient perspectives offer an insight into how novel TVT is received by people living with long term neurological conditions.

### Objective

To explore experiences of people with Parkinson's in using technology as part of a home-based visuo-cognitive training programme compared to traditional approaches to rehabilitation.

### Methods

Eight people with Parkinson's who took part in a pilot randomised cross-over trial, investigating the efficacy and feasibility of TVT compared to standard care, were interviewed to explore their experiences of each arm of the training they received. Integration of Normalisation Process Theory (NPT) into the analysis enabled examination of the potential to embed novel TVT into a home-based rehabilitation intervention for people with Parkinson's disease.

### Results

Three key themes emerged from the thematic analysis as factors influencing the implementation potential of TVT for people with Parkinson's disease: perceived value of technology, perceived ease of use and support mechanisms. Further examination of the data through

anonymized. The ethical approval process was dependent upon no raw data.

**Funding:** This study is funded by a Northumbria University PhD studentship (JD) in collaboration with Senaptec Inc. (Beaverton, Oregon, USA) (PI: SS) and a small British Geriatric Society Movement Disorder Special Interest Group research grant. SS is supported, in part, by a Parkinson's Foundation Post-Doctoral Fellowship for Basic Scientists (PF-FBS-1898-18-21) and a Clinical Research Award (PF-CRA-2073). There was no additional external funding received for this study. the funders had no role in study design, data collection and analysis, decision to publish, or preparation of the manuscript.

**Competing interests:** The authors have declared that no competing interests exist.

the lens of NPT revealed that the implantation and embedding of novel technology was dependent on positive user experience, individual disease manifestation and engagement with a professional.

## Conclusions

Our findings provide insights into the challenges of engaging with technology-based interventions while living with a progressive and fluctuating disease. When implementing technology-based interventions for people with Parkinson's, we recommend that patients and clinicians collaborate to determine whether the technology fits the capacity, preference, and treatment needs of the individual patient.

## Introduction

Parkinson's disease (PD) is a common neurodegenerative condition resulting in a range of motor and non-motor symptoms [1]. Visual and cognitive impairments are common non-motor symptoms, and relate to balance and gait impairment, as well as increased falls risk and reduced quality of life [2–4]. Vision and cognition are interrelated (termed visuo-cognition) [4], which makes treatment complex for people with PD and has led to visuo-cognitive issues being frequently under treated [5].

Technology-based interventions (e.g., mobile applications and stroboscopic glasses) are transforming the landscape of clinical practice and are increasingly being used to deliver visuo-cognitive training that replaces conventional pen and paper or 'therapy game' based interventions [6–11]. Technological visuo-cognitive training (TVT) programmes, combining multiple digital technologies, offer a means of targeting the interdependent visual, cognitive and motor systems which decline in long-term neurological conditions such as PD. TVT could increase the challenge of standard balance and gait exercises (via progressive intermittent visual occlusion with stroboscopic glasses) and deliver a personalised training programme (via mobile applications) that can address visual, sensorimotor and cognitive skills in a home environment [12, 13]. (For full information on the TVT intervention described in this study, please see our previously published study protocol [10]).

Current literature demonstrates that gaps exist between the development of technology-based interventions, such as TVT, and the feasibility of implementation within clinical populations [6, 14–17]. These gaps are in part due to the digital immaturity of healthcare systems, but are also a result of limitations in "real-world" research where technological solutions are developed without consideration of how they will be received by service-users [16]. Acceptability is therefore a key factor in the design and implementation of novel, technological-based healthcare interventions and patient perspectives can provide a useful evaluation tool for new technologies that have limited clinical data to support their effectiveness [8, 18, 19]. Understanding the complexity associated with the implementation of novel interventions entails consideration not just of the effectiveness of the treatment, but also aspects of the approach and context in which it is delivered [17, 20]. A greater role for theoretical approaches in implementation-focussed research is one proposed strategy for overcoming the gaps between new technology development and clinical usability [21].

Normalisation Process Theory (NPT) is a model which provides a conceptual framework to explain the processes by which new health technologies and other complex interventions are routinely embedded and integrated into existing practice [22, 23]. NPT has its origins in

research on the implementation of complex healthcare interventions, so it is not specifically concerned with the specifics of the technology as a tool in itself, but rather as an ensemble of beliefs, behaviours, and practices that may play out differently depending on context [22]. The framework was designed to be applied flexibly, so that it can be used at one or more points in a qualitative study, and it explains how the work of enacting a multimodal intervention (here, the components of TVT) is accomplished through the operation of four mechanisms: coherence (sense-making work), cognitive participation (relationship work), collective action (enacting work), and reflexive monitoring (appraisal work).

Future implementation potential is at the heart of NPT, as the process guides researchers to problem-solve any implementation issues from the initial stages of testing intervention concepts to the latter stages of intervention implementation [24]. Systematic review findings suggest NTP is best applied in healthcare research when the development of interventions is supported by a community of practice and considered legitimate by potential users (e.g., practitioners, patients and carers) [25, 26].

We propose that TVT has the potential to provide an alternative approach to addressing visuo-cognitive dysfunction in people with PD. However, while user acceptance behaviour has been studied for wearable sensors and digital monitoring technologies [27–30], limited attention has been given to the study of factors affecting the integration of technology into rehabilitation interventions for people with PD in their own homes.

Therefore, the aim of this pilot study was to explore the experience of using novel TVT as part of a home-based rehabilitation intervention for people with PD. NPT was used as a analytic framework to facilitate the quality and rigor of the coding and thematic analysis process [31].

## Methods

### Participants & recruitment

Participants in this pilot study were recruited from a cross-over subset of a larger single-centre pilot randomised controlled trial exploring the feasibility and effectiveness of TVT compared with standard care, in people with PD [10]. The main trial started off as a crossover intervention. Eight people with PD completed the crossover interventions before the research team made the decision to convert to a parallel group design due to logistical issues around recruitment and intervention delivery during the Covid-19 period. All eight participants who took part in the preliminary cross-over intervention agreed to be interviewed.

Participants were recruited from Movement Disorders Clinics at the Northumbria Healthcare NHS Foundation Trust, Parkinson's UK Research Support Network, and Dementias & Neurodegenerative Diseases Research Network. Eligibility for the trial meant that study participants had a confirmed diagnosis of PD, were aged over 50 years, living independently, and able to walk and stand without assistance. Exclusion criteria included a history of epilepsy, seizures, migraines, severe motion sickness or sensitivity to light (due to the use of stroboscopic glasses). In addition, individuals with a history of depressive disorder, diagnosis of dementia, or other severe cognitive impairment which would mean they were unable to comply with testing procedures, were not recruited. Individuals with any acute musculoskeletal conditions, unstable medical conditions, or any neurological disorder other than PD were also not eligible.

### Ethical approval and registration

The design of this study conforms to the principles outlined in the Declaration of Helsinki and was approved by the South Central-Berkshire B Research Ethics Committee on 31 March 2021 (ref 21/SC/0042) and Northumbria University's ethical approval system (Ref: 27828). All

participants gave written informed consent and were informed about all relevant aspects of this study, including interviews, before commencing the study. This trial was listed on the ISRCTN registry with study ID ISRCTN46164906 on 21 April 2021.

## Study design

In the study we sought to understand the experiences of people with PD in using technology as part of a home-based rehabilitation intervention, recognizing that the results were best understood when set in the participants' social and experiential contexts. This relativist ontological stance [32] was chosen to reflect the research team's belief that each individual with PD is likely to experience the technology differently and that this experience is best understood by considering contextual factors such as their rehabilitation expectations, prior attitudes toward technology and demographics such as age and severity stage of PD. Semi-structured interviews were chosen to investigate the experiences of participants in the cross-over trial because they facilitate in-depth exploration of these contextual perspectives and therefore provide qualitative data to augment the findings of the ongoing parallel trial.

## Visuo-cognitive training rehabilitation protocol

The overarching trial is described in more detail elsewhere [10], but Fig 1 provides a brief description of the study using the Template for Intervention Description and Replication (TIDieR) framework [33]. In brief, each visuo-cognitive training intervention (i.e., TVT or standard care) consisted of two 45–60 minute training sessions per week for 4 weeks undertaken in participants' homes. Sessions included 20 minutes of visuo-cognitive training exercises and up to 20 minutes of visuo-motor exercises (throwing/catching and balance activities).

## Data collection

Individual face-to-face interviews were conducted between September and December 2021 by investigator (JD) in participant homes (except for in the case of one participant who chose to be interviewed at the Gait Laboratory at Coach Lane Campus, Newcastle upon Tyne due to personal circumstances) following their final training intervention. The interviewer was an experienced physiotherapist with training in qualitative research methods and was known to the eight participants from the intervention period.

The interview schedule used open-ended questions, with no right or wrong answers, to initiate discussions around the chosen topics (Table 1). This approach to the interviews was designed to create an interactive relationship between the researcher and participant, where the multiple potential realities of their experience were explored to produce a detailed and rich data set. All interviews were audio-recorded with consent, anonymised and transcribed verbatim by JD.

## Data analysis

To understand the experiences of people with PD in using technology as part of their rehabilitation, we first employed an inductive thematic analysis [34]. Transcript data were read initially by 2 investigators with experience in qualitative research methods and interviewing (JD and GB) separately to identify participants' attitudes towards the interventions they had received. Ideas and patterns were identified, coded (S1 File) and then grouped according to key themes (Table 2) [34].

| TIDieR component | Description |
|---|---|
| Why (rationale) | Visual, cognitive and motor systems all decline with PD pathology and are difficult to treat. Combining multiple digital technologies (i.e., stroboscopic glasses, mobile applications) into a multi-modal technological visuo-cognitive training (TVT) programme may provide effective rehabilitation for people with PD. It is not yet known whether these approaches are effective or how they compare to standard (traditional) approaches to rehabilitation. |
| What (materials) | **TVT Intervention**<br>• Stroboscopic glasses: Senaptec Strobe Training Eyewear with lenses that flicker between clear and opaque (https://senaptec.com/products/senaptec-strobe)<br>• Sensory training application (https://senaptec.com/products/senaptec-app) delivered via tablet: Apple iPad 7th Gen 10.2 32GB<br><br><br><br>**Standard Intervention**<br>• Pen/paper-based activities adapted from commonly used visuo-cognitive training activities used in clinical practice.<br>• Commercially available games were used to offer comparable challenges to the app-based training drills.<br><br> |
| What (procedures) | A selection of visuo-cognitive training activities delivered via mobile application or pen/paper/game-based activities (approx. 15-20 mins) plus throwing, catching, balance exercises delivered with and without intermittent visual occlusion (up to 20 mins per session). |
| Who provided | A research physiotherapist with specialist neurorehabilitation skills and over 15 years clinical experience alongside experience of conducting qualitative interviews with people with PD and older adults. |
| How (delivery) | Two TVT sessions per week for 4 weeks + two standard visuo-cognitive training sessions per week for 4 weeks. All sessions were delivered and supervised by the provider above. |
| Where | Participants' homes (and one in University Gait Laboratory due to personal circumstance) |
| When and how much | Sixteen x 45-60 minute sessions (two per week for 4 weeks for each arm of intervention) |
| Tailoring | **TVT Intervention**<br>• Throwing/catching/balance activities were delivered with intermittent visual occlusion via strobe glasses (in sitting/standing depending on participant ability and exercise tolerance). Challenge was progressed by increasing length of opaque phase on stroboscopic glasses so that participants had to work with less visual information as well as increasing speed/intensity, duration or amount of each exercise.<br>• App-based drills were programmed to automatically increase in difficulty as visuo-motor skills improve.<br>**Standard Intervention**<br>• The same throwing/catching/balance activities were delivered with uninterrupted vision (in sitting/standing depending on participant ability and exercise tolerance). Challenge was progressed by increasing speed/intensity, duration or amount of each exercise.<br>• Pen/paper and game-based activities were progressed by increasing the challenge/complexity level of tasks such as Tangrams/shape matching, adding increased rate of metronome beats to visual naming tasks, and counting number of errors (e.g., on buzz wire) or adding time constraints/time goals to completion. |
| Modifications | The study design was changed to a parallel-group design for pragmatic reasons. This paper reports on data generated from interviews with the first 8 participants who completed the crossover study. |
| How well (planned) | The provider was present for every home-based intervention and documented session content for each visit, including nature of exercises, level of strobe setting, and length of training tolerated. The provider also made written notes about participant response to exercises/activities, level of engagement and any feedback given by participants in relation to the training. |

**Fig 1. Description of the visuo-cognitive training crossover trail using TIDieR framework.**

Following the thematic analysis, the transcripts and the themes were reread to examine whether and how the themes aligned with the four constructs of NPT: coherence, cognitive participation, collective action and reflexive monitoring [22]. Authors JD and GB conducted

**Table 1. Interview topic guide.**

| Topic area | Discussion points |
|---|---|
| Previous experience | • Problems with vision that may be related to PD.<br>• Previous input from health care team?<br>• Prior experience of treatment for visual problems. |
| Expectations | • Expectations of the interventions before starting the study? |
| Differences between TVT and standard therapy | • Perceived differences between the technological visual training and the standard intervention? |
| Supervision | • Need for professional help or support to use the of technology?<br>• Willingness to continue use of devices at home independently once familiar with the set up and functions |
| Adherence | • Discuss length of training on the technology.<br>• Likelihood of using app/strobe glasses independently in the future. |
| Barriers/facilitators | • Aspects of the mobile app or strobe glasses that made it difficult or easier to engage with. |
| Effects of training (reflection) | • Explore any changes to vision/daily life/mood as a result of the training. |

this analysis individually, and then discussed findings over several meetings. To enhance credibility and trustworthiness of the work, all themes were reviewed and refined by the other members of the research team (RW, RV, RM, SS). These investigators come from a variety of backgrounds, all with experience of research and/or clinical practice in PD and holding different perspectives on the role of technology in rehabilitation. Discrepancies and new ideas were discussed at each stage of the analysis until a consensus was reached. Table 3 sets out how the NPT constructs aligned with the initial thematic analysis.

**Table 2. Key themes and sub-themes derived through thematic analysis of interview data.**

| Key themes | Sub themes |
|---|---|
| **Value of technology** | Context |
| | Understanding impact of visual problems |
| | Expectations |
| | Remote health |
| | "Buy-in" |
| | Compatibility (how the technology fits with existing behaviour) |
| | Physical outcomes |
| | Impact on mood/energy levels |
| | Relative advantage (the superiority of one intervention over the other) |
| | Motivation |
| | Responsiveness |
| | Added challenge |
| | Sense of achievement |
| **Ease of use** | Sociodemographic factors (e.g., age) |
| | Physical factors (relating to PD) |
| | Technical challenges |
| | Technology self-efficacy |
| | Barriers |
| **Support** | Value of face-to-face contact with therapist |
| | Safety |
| | Intention |
| **Study effects** | Behaviour change |
| | Willingness to take part in future research |

**Table 3. NPT constructs aligned with thematic analysis.**

| Construct | Description | Key findings | Linked to theme |
|---|---|---|---|
| **Coherence**<br>The *sense-making* processes that people go through when introduced to a new technological innovation. | **Specification**: do people with PD understand the aims and expected benefits of the technology?<br>**Internalisation**: how do people with PD make sense of the technology (strobe glasses and app) based on lived experience and viewpoint.<br>**Differentiation**: do people with PD distinguish technology-based interventions from more traditional (e.g., pen and paper) approaches to visuo-cognitive training? | The majority of participants had limited awareness or experience of visuo-cognitive problems associated with PD and therefore entered the study with a fairly vague understanding of how the technology might impact them or what the intervention might involve in the context of their personal experience.<br>*"Well I was interested to see... erm, what the technology would show. And... and, you know, having had Parkinson's for 5 years, how I would compare to other people."* [PD01]<br>Sense-making work occurred through the process of differentiation as participants had the opportunity to compare the two interventions based on lived experience and perceived outcomes. | Value of technology |
| **Cognitive participation**<br>The *relational work* needed to implement and sustain use of the technology. | **Legitimation**: do people with PD believe that the technology is right for them; do they "buy in"?<br>**Enrolment**: who do people with Parkinson's engage with to provide support?<br>**Activation**: will people with PD continue to use the technology?<br>**Initiation**: will people with PD drive the implementation of the intervention? | Participants expressed mixed attitudes towards the use of technology based on sociodemographic factors such as age and physical capabilities. However, even for some self-proclaimed "technophobes", there was still "buy-in" to the technological intervention.<br>Support mechanisms were also raised as a factor likely to influence ongoing use of the technology.<br>Participants expressed intention to use the technology if efficacy was shown.<br>Intention to continue use was also related to technology self-efficacy.<br>Participants recognised that the technology added a level of difficulty/challenge to the intervention which could prove to be a barrier to implementation in people with more advanced PD. | Ease of use<br>Value of technology<br>Support mechanisms |
| **Collective action**<br>The *enacting work* that is required for the technology to work in practice. | **Skill set workability**–how does the technology work in practice? | There were mixed views about ease of use and interactivity of the technology which impacted on how the technology worked for people with PD in practice. Some participants reported challenges while others remarked on the simplicity and ease of use of the technology.<br>*"The thing I find with the screen, and it's this thing I have with me iPad and all that, I have very greasy fingers, right. So sometimes when I was touching the screen it was as if it wasn't picking up"* [PD08]<br>*"I think the app is probably easier and simpler to use and you get that visual... you know you're doing better... You get that it's encouragement."* [PD06] | Ease of use |
| **Reflexive monitoring:**<br>The *evaluation work* that people do to assess and understand the ways that the technology affects them. | **Individual appraisal**: how do people with PD assess the value of the technological intervention? | Physical outcomes, impact on mood/energy levels and motivational factors influenced participants' appraisal of the technology.<br>*"No, I didn't think I was tired [after TVT] ... I think I was... if I was anything it woke me up... it stimulated you to ... to go on for the rest of the day."* [PD01]<br>For a number of participants, the value of the technology was closely linked to the value placed on the face-to-face contact with the researcher.<br>*"I've got a link to a Parkinson's nurse, but I've never used it... and just that regularity, I guess, of doing things. You coming to my home and making me think about it has been really good."* [PD07] | Value of technology<br>Support mechanisms |

## Results

### Respondent characteristics

Participant characteristics at study enrolment are shown in Table 4. The main body of results are presented using the NPT concepts of *coherence*, *cognitive participation*, *collective action* and *reflexive monitoring*. Data are presented to support analysis and labelled by identifier number (PD01 = participant 1).

### Coherence

The first NPT construct–coherence–describes how people with PD make sense of the technology in relation to their condition and is linked to the theme "*value of technology*". If people with PD do not perceive the use of technology as relevant to them and their condition, they may not fully engage with it.

This study explored the use of novel technology to address visuo-cognitive function, an area which often receives less clinical attention due to the focus on motor manifestations in PD. This is reflected in the data, as half the participants were "totally unaware" that PD could cause visual dysfunction and did not link their visual problems to PD:

> "*I've got I've eye problems and I've got iritis which is inflammation of the iris. Then, just be about a few weeks before lockdown I had, erm. . . detached retina. So, I have had discussions about my eye. . . Totally separate to Parkinson's.*" PD03

Some participants struggled to differentiate between the effects of ageing and their condition, meaning they had not discussed the problems with their PD team:

> "*. . . there is parts of the day when my eyesight is worse than it would be, so, once the light starts to get a bit dim. . . erm. . .. My vision seems to get less clear. . . Apart from that, I don't know whether that's Parkinson's or whether that's just old age. But in fading light, my eyesight's not as good as it is in strong sunlight.*" PD05

Unsurprisingly, given this lack of context, six out of the eight participants entered the study with little expectation of what the visuo-cognitive training interventions might involve and how the technology might be of value to them and their condition:

**Table 4. Participant characteristics at study enrolment.**

|  | Gender | Age (yrs) | H&Y[a] (1–5)[b] | Disease duration (yrs) | MoCA[c] (0–30)[d] | MDS-UPDRS III[e] (0–132)[a] | Living status | Living area | Years of education |
|---|---|---|---|---|---|---|---|---|---|
| **PD01** | Male | 66 | 2 | 5 | 26 | 54 | With partner | Urban | 10 |
| **PD02** | Female | 64 | 3 | 20 | 26 | 49 | With partner | Urban | 9 |
| **PD03** | Male | 59 | 1 | 4 | 30 | 13 | With partner | Urban | 16 |
| **PD04** | Male | 57 | 3 | 4 | 29 | 43 | With partner | Urban | 12 |
| **PD05** | Male | 56 | 1 | 6 | 30 | 15 | With partner | Urban | 13 |
| **PD06** | Male | 58 | 1 | 2 | 30 | 19 | With partner | Urban | 11 |
| **PD07** | Male | 62 | 2 | 1 | 29 | 24 | With partner | Urban | 18 |
| **PD08** | Male | 67 | 1 | >1 | 29 | 14 | With partner | Urban | 11 |

[a]H&Y: Hoehn & Yahr scale

[b]Lower scores are better

[c]MoCA: Montreal Cognitive assessment

[d] Higher scores are better

[e]MDS-UPDRS III: Movement Disorder Society Unified Parkinson's Disease Rating Scale Part 3 Motor section

*"I had no idea about what . . . you were going to do. . . I was just happy to volunteer thinking that if my input helps other people or myself, I get something out of it. Well, then it's a bonus."* PD06

One participant recognised the potential value of technology in relation to remote delivery and engaging with the wider PD community:

*"Technology has got a place. . . because the lack of resources. . . technology has always got to be the way forward. Because you cannot always have somebody sitting with you. You know, measuring what you do. And. . . I would imagine remotely, you know that at some point in time there could be a thousand people with Parkinson's all logged into a computer, and. . . not playing against each other, but recording data. . ."* PD01

Another participant even alluded to an association between technology usage and a way of engaging in society:

*"It's this actually helps us feel like we're doing something. That we're integrating into normal life"* PD03

Further "sense-making" work took place as participants compared their experiences of the standard training with the TVT. A summary of participant preferences is presented in Table 5 which demonstrates there was mixed opinion in relation to the value placed in the technology-based interventions and highlights the difficulty some participants had in differentiating between the two.

**Table 5. Participant preferences.**

| | Preference | Supporting quotation |
|---|---|---|
| PD01 | Technology | *"If I was being left to do it on my own. Then the app would be easier, and you would get instantaneous responses to it. If you're doing paper exercises, it's like handing in your exam results in at school and you're waiting for the result to come back. You would get an immediate response from the app that said: '30 seconds and tomorrow it was 29 seconds and. . .' and you would see the improvement."* |
| PD02 | Undecided | *"There was something in each one that I preferred. . . Err. . . In the one I've done. . . the last part, I enjoyed them because them were the sort of things I work on- on the computer"* |
| | | *The games and the paper based. . .I just- I just thoroughly enjoyed. . . it when I've done it. Err. . . I think it's–It's something I've always enjoyed, that sort of thing. Erm, er. . . it was just a carry on from what I do at home anyway. . ."* |
| PD03 | Standard | *I'd rather sit through the paper-based stuff. . . But I have a slight aversion to tech. . . It's just 'cause I work with it for so long it's. . . when I go home, I . . . Just switch off from it."* |
| PD04 | Technology | *"Definitely the app because my. . . Parkinson's manifestation, if you like, one of the problems is writing. Er. . . and just holding a pen. I've got to concentrate very much in writing a letter or anything like that, so that'll be much harder for me."* |
| PD05 | Technology | *"I mean they were both good. I think if I had to choose, I would go for the technology one first. Because I think there's probably more benefit from that one."* |
| PD06 | Undecided | *"The electronic things that you did were also very good, and I honestly felt that I benefited from both of them. I could feel a marked improvement doing both of them each time we did it. . . So, I was happy to do either of them. I personally have no great choice or preference between the two because I could feel myself improving with both of them."* |
| PD07 | Undecided | *"I honestly don't think I have a preference. Umm, I've quite enjoyed the little things on the iPad, but I did like the tangrams and the blocks and things as well. So genuinely no, no preference. If you were to say we're doing another four weeks and you've been chosen for one or the other, it wouldn't worry me."* |
| PD08 | Standard | *". . . . But I think that might be because of my age. I'm still more comfortable with paper things."* |

## Cognitive participation

Our data suggest that the concept of technology-based interventions held a degree of *coherence* for a minority of participants at the start of the study, but the sense-making process derived through lived experience of both interventions lead to increased value being placed in the technology-based intervention by a greater number of participants.

The second NPT construct–cognitive participation–adds context to this sense-making work. This construct draws on data from all three themes—*perceived ease of use, value of technology and support mechanisms*—to understand the relational work needed to be undertaken by people with PD to implement and sustain use of the technology. It involves understanding the extent to which participants believe technology is right for them, and people with PD in general, to use and how much they "buy into" the intervention (*legitimation*). It explores engagement with others (*enrolment*) in connection with the work required for people with PD to sustain the use of the technology beyond participation in the study (activation), and what is required to drive its' implementation (*initiation*).

Participants expressed mixed attitudes towards the use of technology within the intervention. Some referred to individual characteristics such as their age as an influencing factor in the context of their decision making whereas others expressed uncertainty about technology, based on personal experience (Table 5). These data demonstrate that pre-existing attitudes did not necessarily affect participants' 'buy in' to the technology which proved to be an unexpected outcome for some participants:

> "*I'd go for the tech. Normally I'm a Luddite, which is a bit of surprise, but yeah, I would. I'd go for the tech.*" PD04

The need to enrol support (both from family members and health professionals) was recognised as an important factor in facilitating ongoing use of the technology at home. This was associated with the need for technical help with settings as well as safety considerations when using the strobe glasses:

> "*. . . as long as you have somebody with you. It wouldn't necessarily need to be a professional. Just as long as you had somebody with you where you didn't end up falling over because you've set it too high*" PD01

Participants also recognised that the technology added a degree of difficulty or challenge to the intervention which could prove to be a barrier to implementation in people with more advanced PD:

> "*Not [need help] from a technical point of view, I would say no. I do know. . . I felt fine wearing the glasses and I felt I could cope with them if I wore them at home. But I don't know if possibly somebody who had more advanced Parkinson's might be more at risk wearing the glasses.*" PD03

Furthermore, technology self-efficacy also featured as a significant determiner of ongoing use of the technology at home. Concerns about encountering technical issues meant that some participants would be less likely to maintain use of the technology beyond the study period desperate their preference for the technological intervention:

> "*I probably enjoyed the app more but I'd probably do the paper ones more easily because. . . I'm a bit of a technophobe. So if the app starts playing up, or I can't get connected to the*

*internet. . . then I'll think, well I can't be bothered to do it. Whereas if I've just got the paper exercises. . . pieces of paper in front of you, you can get the piece of paper out or a book of tests or whatever, you can just work your way through it"* PD05

Irrespective of the value placed in the technology and the perceived need for support, participants still expressed intention to use the devices if efficacy was shown. This is exemplified by the following quotation from PD03 who previously stated an aversion to technology:

"*Oh, I'd definitely use it if there was some benefit.*" [PD03]

Participants also described new motivation around behaviour change in relation to technology-based "brain training" type activities as a result of their introduction to, and experience with, TVT:

"*I definitely think I'm going to try and do something. Whether it's find some brain training exercises to do, some games on my phone. . . to keep my brain active.*". PD05

## Collective action

The third NPT construct–collective action–addresses how the technology works in practice and is closely linked to the theme "*ease of use*" as it explores the levers and barriers to people with PD engaging with the technology.

Overall, few technical issues arose during the TVT, and those that did occur (such as temporary disturbances in Wi-Fi connectivity causing the app to drop out, or adjustments to the stroboscopic settings on the glasses), were quickly overcome:

"*No, I mean we didn't have any problems, or I think we had one little Wi-Fi down, but generally it behaved itself and once I understood the instructions for the games, that was good*" PD07

There were mixed views about the interactivity of the technology. Some participants remarked that the simplicity of the technology was a significant lever to its utilisation, whereas others reported challenges with the intervention, notably with respect to interactions with the mobile application touch screen tablet device:

"*And also, the tech I think had trouble sometimes when there was touch sensitive screens. I dunno if it was me or. . . or the tech, but it wasn't always reacting, which was a little bit frustrating. . .*" PD03

The immediacy of feedback provided by the technology was seen as an advantage over the standard care intervention and emerged as a significant motivator for participants to continue to engage with the technology independently (see Table 5).

Physical limitations associated with PD were discussed in relation to how the technology worked in practice and its perceived advantages over pen and paper type activities:

"*So, using the app. . . There won't be as many kind of physical problems as is when you use the pen and paper and the shapes, and in particular the buzz wire thing. . . Now that, if you have a tremor, that brings that into play a lot. . . whereas your tremor. . . it hasn't affected me at all on the app.*" PD04

## Reflexive monitoring

The last NPT construct—reflexive monitoring—explores the appraisal work undertaken by people with PD based on their experiences of the visuo-cognitive training:

> "*I mean you could say the first (standard), it was just kind of pure fun and the second (TVT) is a combination of fun and quite hard work with the concentrating.*" PD04

There was a general consensus that the addition of technology added a level of difficulty to the training. Some participants remarked positively about the motivational potential of the technology compared to the standard training:

> "*. . . I felt it's always getting faster, and I think it's the encouragement of the way the app sort of leads to a competitive instinct in me to try and beat the machine. Like a fairground ride or an amusement arcade machine, you want to beat your best score. . . which you don't get off the paper. . .*" PD06

Others reflected on the sense of achievement they felt as a result of engaging in interventions that were deemed more challenging as a result of the technology:

> "*I think you get the standard. Yeah, there is some satisfaction there. But I think when I've got the glasses on . . . I'm trying to keep my balance and do the little exercise. It's more the achievement satisfaction there. . . 'cause you're doing something that is more difficult.*" PD04

For some participants however, the added challenge imposed by the technology was perceived as a disincentive which detracted from their participation in the training:

> "*I would prefer the exercises without the glasses, yes. . . probably not best for the training, but they're easier to do, easier to complete. . .*" PD05

Despite the mixed appraisal, most people with PD reported psychological improvements as a result of engaging with the technology. For some, this presented as increased mood and alertness whilst for others this manifested as an increased confidence in physical functioning:

> "*It seemed to. . . awaken parts of your brain, you know, that you weren't aware were asleep. . . No, I didn't think I was tired* [after the TVT]. . . *if I was anything it woke me up. . . it stimulated you to. . . to go on for the rest of the day*" PD01

Participants also reflected very positively on the patient-therapist interaction that underpinned both arms of the intervention. It became evident that for some participants, the inherent value they placed in the face-to-contact with the researcher underpinned their evaluation of both interventions:

> "*It's been an eye-opener because my diagnosis was right at the beginning of lockdown, and I've not really had access to expertise like yours, the physiotherapy expertise and just the knowledge that you've got. It's been the first block of time where somebody's been giving me feedback about my specific symptoms. . .. and I've got a link to a Parkinson's nurse, but I've never used it. . . and just that regularity, I guess, of doing things. You coming to my home and making me think about it has been really good.*" PD07

These data are particularly pertinent as they add context to the sense-making, relational and enacting work in the previous three NPT constructs and emphasise the value of the patient-therapist interaction in facilitating the implementation and integration of novel technology into a home-based rehabilitation programme.

### Data outside the coding framework

A small proportion of data identified during the thematic analysis fell outside the NPT framework and is categorised under the theme "study effects" (Table 2). These data either (1) were focused on behavioural change unrelated to technology usage or (2) willingness to participate in future research. The following quotation exemplifies data of this nature:

> "*it's been very interesting, sort of thing, you know, and I'm quite willing to take part in. . . in any kind of future [research].*"

## Discussion

The purpose of this study was not to determine the efficacy of TVT compared to standard care, but rather to explore what aspects of the technological interventions contribute to their appeal and usability in people with PD. Conceptualising the themes generated from the interviews through the lens of NPT has enhanced our understanding and interpretation of factors involved in implementing and embedding technology into home-based interventions from the perspectives of people with PD. Our findings supplement previous studies that have investigated perceptions and attitudes of older adults towards new technologies [35, 36].

The first NPT construct of *Coherence* refers to the sense-making work undertaken when a new technological intervention is implemented. It determines whether users understand how technology will affect them personally, whether they have a shared view of its purpose and whether they see it as differing from existing or "standard" practices [37].

Our findings suggest that people with PD had few expectations of what value the technology might add to their rehabilitation experience over the standard care intervention prior to entering the study. Current literature suggests that in order to evaluate a novel technology, consideration must be given to the health need that the technology is intended to address [38]. In the field of PD, visuo-cognitive impairments remain under-recognised in research as well as in clinical practice [39] which may explain why participants lacked awareness of how technology might benefit them personally in the context of visual or cognitive dysfunction.

Since the emergence of Covid-19, there is a wealth of literature purporting the more generic benefits of technology-enabled remote healthcare [40] but this was not reflected in the sense-making work undertaken by participants in this study. Despite the fact that our study took place during the pandemic, only one participant (PD01) made reference to the value of technology in terms of minimising in-person visits and reducing face-to-face contact with health professionals. The novelty of the technology in comparison to the standard care intervention fostered interest among the participants in relation to how it might benefit themselves or other people with PD, rather than them perceiving it's value in terms of wider service delivery. Interestingly, another participant viewed technology usage as a way of positively engaging with society and "integrating into normal life" which is a notion supported by other studies looking at negative ageing stereotypes associated with non-use of technology [41].

Participants were better able to differentiate between the technology and standard care treatments on completion of the study, having had the opportunity to experience, and therefore make sense of, both interventions. Adding context to this sense-making work, the second

NPT construct of *Cognitive Participation* integrated data from all three themes and revealed a combination of factors influencing the relational work needed to implement and sustain use of this novel technology.

On completion of the training, technology 'buy in' was apparent, even for those participants who denied having visuo-cognitive problems, or those who were self-proclaimed "techno-phobes". This response is in keeping with past studies, which have shown that the perceived value of a technological intervention appears to have a greater influence on the degree of legitimation (or "buy-in") than individual characteristics and personal attitudes to technology [35, 36, 42]. These findings help us to understand that work is required to ensure that people with PD understand the meaning, utility and relevance of a novel technological intervention such as TVT in order for its implementation to be optimised in practice. People with PD benefit from participatory experience of a novel technology for it to hold coherence for them in the context of their rehabilitation.

People with PD recognised that enrolment of support from family members and health professionals was necessary for the successful implementation of TVT. The added degree of difficulty associated with the technology-based intervention was identified as a potential barrier to implementation for people with more advanced disease. This finding is in keeping with literature exploring perceived barriers to exercise which identify safety concerns (such as fear of falling) as a major barrier to engaging in interventions for people with PD [27, 43].

The third NPT construct, *Collective Action*, predominantly draws on data from the theme "ease of use" and refers to the work required to implement the technology in practice. Our findings showed there were mixed views about ease of use and interactivity of the technology which impacted on how the technology worked for people with PD.

Technical challenges, such as difficulty engaging with the touch screen interface on the device, did lead to frustration and dissatisfaction for some participants. Similar results have been found in previous studies where loss of power and fine motor ability can represent a barrier to users with long-term health conditions [44, 45]. Conversely, some people with PD found the technology more accessible than the standard interventions when taking into account physical limitations, such as tremor. This finding suggests that confidence in technology among people with PD is in part related to their ability to physically engage with the technology. This confidence may fluctuate given the variability of symptoms associated with PD [46]. Consideration must therefore be given to offering alternative, multi-modal or 'combination' approaches to allow people with PD to engage with interventions in accordance with their fluctuating motor and non-motor symptoms.

There was a consensus that the technology offered an immediacy of feedback which were seen as an advantage over the standard care intervention. Parker et. al. suggest that elements of feedback such as rewards or knowledge of results are required to facilitate usage [47] and the feedback provided by the technology (both in the form of result scores on the app and operational levels on the strobe glasses) emerged as a significant motivator for participants to continue to engage with the technology in this study.

The final NPT construct, *Reflexive Monitoring*, explores the appraisal work undertaken by people with PD in relation to their experiences of the visuo-cognitive training and encompasses data from the themes *value of technology* and *support mechanisms*. The cross-over nature of the study adds weight to our data because participants were able to compare and reflect on the perceived benefits of the two modes of training delivery. As discussed previously, there was a general consensus that the addition of technology added a level of difficulty to the training which was not apparent in the standard care intervention. However, this was appraised differently according to how each participant perceived the value of the added challenge. While some remarked positively on the sense of achievement they felt using the

technology, the added difficulty was perceived as a disincentive by other participants because it detracted from their participation in the intervention. Similar views were expressed by participants in the study by Roswell et. al., who reported mixed reactions to the use of equipment to increase the demands of an exercise intervention for people with PD [48]. Our findings support the notion that clear explanations and descriptions of how the technology might impact performance could facilitate a greater acceptance of the technology and lead to a more positive appraisal of its effects [48].

Also evident in the appraisal work undertaken by participants was a reflection on the psychological improvements associated with technology usage. Increased mood and alertness were frequently mentioned as outcomes of the TVT with words such as "buzz" and "boost" used to describe its effects. The responsiveness and stimulation of the technology may therefore prove to be of significant motivational value for people with PD to help them overcome inherent obstacles to rehabilitation in the form on/off fluctuations, co-occurring depression, anxiety and apathy [49, 50].

One of the most pertinent findings from the study was the value that people with PD placed in the therapy interaction that accompanied the training sessions. The study took place in 2021 while the UK was still experiencing social restrictions due to the Covid-19 pandemic [51]. Participants commented on the lack of contact from their healthcare providers in the 12 months preceding their participation in the study, and it may be that the value they placed on the face-to-face contact with the researcher was heightened as a result of the pandemic. However, previous studies in elderly populations have reported similar findings, with individuals valuing the human interaction that accompanies technology more than the technology itself [52]. Not only does this highlight the importance of introducing technology-based interventions in a supported context, it also demonstrates that their use will not automatically reduce the need for direct clinical contact to maximise adherence to rehabilitation.

## Strengths and limitations

This is the first qualitative study to explore people with PD experiences and attitudes towards visuo-cognitive rehabilitation using novel technology versus more traditional approaches. There were a number of strengths of this study. First, the cross-over design allowed a unique insight into the opinions of people with PD who had the opportunity to experience both approaches to treatment. Second, use of NPT to frame the data analysis allowed a theoretical approach to determining factors that influence the successful implementation of a TVT rehabilitation programme. The authors acknowledge that there is some overlap between the themes and the constructs. However, the application of NPT is designed to be flexible to allow non-linear analysis which helped us understand and add context to the processes of implementation and embedding technology into home-based care for people with PD [53].

There were also several limitations to our study. The same research physiotherapist who delivered the interventions also conducted the interviews. This may have impacted on participant responses in the interviews and therefore there is a risk of bias in the findings. However, this method allows for a thorough understanding of the material as it is through the researcher's facilitative interaction that a context is created where respondents are set at ease and share rich data regarding their experiences [54, 55]. Additionally, this study involved a small sub-sample of a pilot RCT (n = 8), and females were under-represented (7M:1F), which mean the findings may not be transferable to other contexts and populations. Furthermore, all participants in the study had access to the internet and were active users of technology either in a work setting or through personal use, so they may not be representative of the wider PD population.

Finally, only people with PD were involved in the interviews. Successful implementation of new technology depends on the acceptability of the intervention from a perspective of all stakeholders and therefore it would be useful to consider clinician feedback in any future studies exploring the use of TVT [14, 56].

## Implications for practice

People with PD represent a heterogenous group with a broad range of individual preferences, experience, and fluctuating needs. Differentiation is important: rather than thinking in terms of blanket application of TVT, clinicians need to understand which patients are likely to be receptive to the technology and ensure alternative options (or combination interventions) are available for people who want them. For example, individuals who work regularly with technology may be inclined to opt for alternative (non-technological) approaches to avoid prolonged screen-use during non-working/leisure time. Others may find technology-based interventions preferable if they struggle with dexterity or pen and paper-based tasks. People with PD are receptive to novel technology but require one-to-one support to ensure they have the confidence to use the tools independently. Face to face interactions and endorsement from a trusted health professional facilitate use of technology and are likely to increase uptake and sustained use within the context of rehabilitation. Future work needs to determine how much influence this face-to-face contact with a health professional has on the integration of technology into rehabilitative practices. A greater 'buy-in' from people with PD can be addressed through more rigorous research and dissemination of the efficacy of technology-based interventions through education and supported use.

## Conclusion

As we look to the future, technological advances will pave the way for new and innovative approaches to rehabilitation practices. Our qualitative findings provide valuable insights into the challenges and benefits of engaging with novel technology while living with a progressive and fluctuating disease. Conceptualising themes through the lens of NPT focussed our understanding of the barriers and facilitators to the adoption and embedding of technology in rehabilitation interventions for people with PD. When considering the implementation of technology-based interventions for people with PD, in light of our findings we recommend that patients and clinicians collaborate to determine whether the technology fits the capacities, preferences, and treatment needs of the individual patient.

## Supporting information

**S1 File. Coding framework.**
(DOCX)

## Acknowledgments

This study is based at the Physiotherapy Innovation Laboratory (Website: www.pi-lab.co.uk, Twitter: @Physio_In_Lab).

## Author Contributions

**Conceptualization:** Julia Das, Gill Barry, Richard Walker, Rodrigo Vitorio, Rosie Morris, Samuel Stuart.

**Data curation:** Julia Das.

**Formal analysis:** Julia Das, Gill Barry, Richard Walker, Rodrigo Vitorio, Rosie Morris, Samuel Stuart.

**Funding acquisition:** Rosie Morris, Samuel Stuart.

**Investigation:** Julia Das.

**Methodology:** Julia Das, Gill Barry, Richard Walker, Rodrigo Vitorio, Rosie Morris, Samuel Stuart.

**Project administration:** Julia Das, Gill Barry, Rodrigo Vitorio, Rosie Morris, Samuel Stuart.

**Supervision:** Gill Barry, Richard Walker, Rodrigo Vitorio, Rosie Morris, Samuel Stuart.

**Writing – original draft:** Julia Das, Gill Barry.

**Writing – review & editing:** Julia Das, Gill Barry, Richard Walker, Rodrigo Vitorio, Rosie Morris, Samuel Stuart.

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
