## [Decision Letter · Decision Letter 0]

8 Feb 2023

PONE-D-22-33798The integration of technology into a home-based visuo-cognitive training intervention for people with Parkinson’s: is the future digital?PLOS ONE

Dear Dr. Stuart,

Thank you for submitting your manuscript to PLOS ONE. After careful consideration, we feel that it has merit but does not fully meet PLOS ONE’s publication criteria as it currently stands. Therefore, we invite you to submit a revised version of the manuscript that addresses the points raised during the review process.

We look forward to receiving your revised manuscript.

Kind regards,

Jeffrey S. Hallam, Ph.D., FRSPH

Academic Editor

PLOS ONE

Journal Requirements:

2. Please note that PLOS ONE has specific guidelines on code sharing for submissions in which author-generated code underpins the findings in the manuscript. In these cases, all author-generated code must be made available without restrictions upon publication of the work. Please review our guidelines at https://journals.plos.org/plosone/s/materials-and-software-sharing#loc-sharing-code 

and ensure that your code is shared in a way that follows best practice and facilitates reproducibility and reuse.

"This study is funded by a Northumbria University PhD studentship (JD) in collaboration with Senaptec Inc. (Beaverton, Oregon, USA) (PI: SS). SS is supported, in part, by a Parkinson’s Foundation Post-Doctoral Fellowship for Basic Scientists (PF-FBS-1898-18-21) and a Clinical Research Award (PF-CRA-2073)."

"This study is funded by a Northumbria University PhD studentship (JD) in collaboration with Senaptec Inc. (Beaverton, Oregon, USA) (PI: SS). SS is supported, in part, by a Parkinson’s Foundation Post-Doctoral Fellowship for Basic Scientists (PF-FBS-1898-18-21) and a Clinical Research Award (PF-CRA-2073). "

7. We note that Figure (Table 1) in your submission contain copyrighted images. All PLOS content is published under the Creative Commons Attribution License (CC BY 4.0), which means that the manuscript, images, and Supporting Information files will be freely available online, and any third party is permitted to access, download, copy, distribute, and use these materials in any way, even commercially, with proper attribution. For more information, see our copyright guidelines: http://journals.plos.org/plosone/s/licenses-and-copyright.

a. You may seek permission from the original copyright holder of Figure (Table 1) to publish the content specifically under the CC BY 4.0 license. 

Reviewers' comments:

Reviewer's Responses to Questions

**Comments to the Author**

1. Is the manuscript technically sound, and do the data support the conclusions?

Reviewer #1: Yes

Reviewer #2: Yes

2. Has the statistical analysis been performed appropriately and rigorously? 

Reviewer #1: N/A

Reviewer #2: N/A

3. Have the authors made all data underlying the findings in their manuscript fully available?

Reviewer #1: No

Reviewer #2: Yes

4. Is the manuscript presented in an intelligible fashion and written in standard English?

Reviewer #1: Yes

Reviewer #2: Yes

5. Review Comments to the Author

Reviewer #1: This paper examined the experiences of individuals with Parkinson's disease while using an at-home visuo-cognitive training program. The authors used Normalization Process Theory to evaluate the subjective experience the individuals had with the program. The primary goal was to evaluate potential factors that would drive use of this program from the end-user perspective. The research subjects were a subset of a larger randomized controlled trial. Overall the paper is well-written and interesting, although is a small sample and preliminary. This initial evaluation of the program/technology is an important part of the developmental process. The authors provided a detailed description of the analysis/theory which is appreciated. Below are some minor suggestions to improve the quality of the manuscript.

Pg. 3, line 67: Although the technology is outlined in Table 1, it would be helpful to add a general overview of the training in the introduction.

Pg. 3, line 68: please define these "gaps", it will help to support your "WHY" this is needed.

Results: There is some repetitions between the tables and the text-- especially in subject quotes. Please make sure that these statements are presented only once.

Although this was a qualitative study, it would be helpful to add some data about trends in subject statements, similar to what was noted on line 200 -- the majority of the subjects (give numbers to support this statement, for example 6 out of 8)

Table 5: Please remove extra lines in this table to make it cleaner.

Overall there are a few typos and grammatical errors- please review for those.

Reviewer #2: I really enjoyed this article - it was a well written qualitative study. I honestly have so few suggestions that I'm almost embarrassed as a reviewer! The only comment I have is just to double check consistency in format of headings - sometimes all words are capitalized, sometimes only the first is. That's such a minor thing, but the one thing I caught. From a methodological standpoint, it was great. The authors noted the limitation of the small sample, but it was also appropriate for the design. Well done, all around.

6. PLOS authors have the option to publish the peer review history of their article (what does this mean?). If published, this will include your full peer review and any attached files.

Reviewer #1: No

Reviewer #2: **Yes: **Meg Patterson

---

## [Author Response · Author response to Decision Letter 0]

27 Feb 2023

Please see "Response to Reviewers" document uploaded on system.

---

## [Decision Letter · Decision Letter 1]

17 Apr 2023

The integration of technology into a home-based visuo-cognitive training intervention for people with Parkinson’s: is the future digital?

PONE-D-22-33798R1

Dear Dr. Stuart,

We’re pleased to inform you that your manuscript has been judged scientifically suitable for publication and will be formally accepted for publication once it meets all outstanding technical requirements.

Kind regards,

Jeffrey S. Hallam, Ph.D., FRSPH

Academic Editor

PLOS ONE

Additional Editor Comments (optional):

Reviewers' comments:

Reviewer's Responses to Questions

**Comments to the Author**

1. If the authors have adequately addressed your comments raised in a previous round of review and you feel that this manuscript is now acceptable for publication, you may indicate that here to bypass the “Comments to the Author” section, enter your conflict of interest statement in the “Confidential to Editor” section, and submit your "Accept" recommendation.

Reviewer #1: All comments have been addressed

Reviewer #2: All comments have been addressed

2. Is the manuscript technically sound, and do the data support the conclusions?

Reviewer #1: Yes

Reviewer #2: Yes

3. Has the statistical analysis been performed appropriately and rigorously? 

Reviewer #1: Yes

Reviewer #2: Yes

4. Have the authors made all data underlying the findings in their manuscript fully available?

Reviewer #1: Yes

Reviewer #2: Yes

5. Is the manuscript presented in an intelligible fashion and written in standard English?

Reviewer #1: Yes

Reviewer #2: Yes

6. Review Comments to the Author

Reviewer #1: Thank you for your consideration of my suggestions. The manuscript is much improved. Congratulations on your good work.

Reviewer #2: (No Response)

7. PLOS authors have the option to publish the peer review history of their article (what does this mean?). If published, this will include your full peer review and any attached files.

Reviewer #1: No

Reviewer #2: No

---

## [Editor Report · Acceptance letter]

20 Apr 2023

PONE-D-22-33798R1 

The integration of technology into a home-based visuo-cognitive training intervention for people with Parkinson’s: is the future digital? 

Dear Dr. Stuart:

I'm pleased to inform you that your manuscript has been deemed suitable for publication in PLOS ONE. Congratulations! Your manuscript is now with our production department. 

Kind regards, 

on behalf of

Dr. Jeffrey S. Hallam 

Academic Editor

PLOS ONE